# Plasticity of cell migration resulting from mechanochemical coupling

**Yuansheng Cao[†], Elisabeth Ghabache[†], Wouter-Jan Rappel***

Department of Physics, University of California, San Diego, La Jolla, United States

**Abstract** Eukaryotic cells can migrate using different modes, ranging from amoeboid-like, during which actin filled protrusions come and go, to keratocyte-like, characterized by a stable morphology and persistent motion. How cells can switch between these modes is not well understood but waves of signaling events are thought to play an important role in these transitions. Here we present a simple two-component biochemical reaction-diffusion model based on relaxation oscillators and couple this to a model for the mechanics of cell deformations. Different migration modes, including amoeboid-like and keratocyte-like, naturally emerge through transitions determined by interactions between biochemical traveling waves, cell mechanics and morphology. The model predictions are explicitly verified by systematically reducing the protrusive force of the actin network in experiments using *Dictyostelium discoideum* cells. Our results indicate the importance of coupling signaling events to cell mechanics and morphology and may be applicable in a wide variety of cell motility systems.

DOI: https://doi.org/10.7554/eLife.48478.001

## Introduction

Eukaryotic cell migration is a fundamental biological process that is essential in development and wound healing and plays a critical role in pathological diseases, including inflammation and cancer metastasis (*Ridley et al., 2003*; *Roussos et al., 2011*; *Montell, 2003*). Cells can migrate using a variety of modes with a range of corresponding morphologies. The repeated extensions and retractions of pseudopods in amoeboid-like cells, for example, result in a constantly changing morphology and random migration while keratocyte-like cells have a stable and broad actin-rich front, a near-constant shape, and move in a persistent fashion (*Webb and Horwitz, 2003*; *Keren et al., 2008*). Furthermore, many cells do not have a unique migration mode and can switch between them, either as a function of the extracellular environment or upon the introduction of a stimulus (*Paul et al., 2017*; *Bergert et al., 2012*; *Charras and Sahai, 2014*; *Liu et al., 2015*; *Petrie and Yamada, 2016*; *Miao et al., 2017*). This plasticity is currently poorly understood and is thought to play a role in pathological and physiological processes that involve cell migration, including cancer metastasis (*Friedl and Alexander, 2011*).

A key step in cell migration is the establishment of an asymmetric and polarized intra-cellular organization where distinct subsets of signaling molecules, including PAR proteins, Rho family GTPases and phosphoinositides, become localized at the front or back of the cell (*Jilkine and Edelstein-Keshet, 2011*; *Rameh and Cantley, 1999*; *Goldstein and Macara, 2007*; *Raftopoulou and Hall, 2004*; *Rappel and Edelstein-Keshet, 2017*). In the absence of directional cues, this symmetry breaking can be a spontaneous and dynamic process with waves of cytoskeletal and signaling components present on the cell cortex (*Vicker, 2002*; *Weiner et al., 2007*; *Whitelam et al., 2009*; *Case and Waterman, 2011*; *Gerisch et al., 2012*; *Allard and Mogilner, 2013*; *Gerhardt et al., 2014*; *Barnhart et al., 2017*). Addressing this spontaneous symmetry breaking and the role of waves have generated numerous theoretical studies (*Jilkine and Edelstein-Keshet, 2011*; *Meinhardt, 1999*; *Otsuji et al., 2007*; *Csikász-Nagy et al., 2008*; *Beta et al., 2008*; *Mori et al., 2008*;

**\*For correspondence:**
rappel@physics.ucsd.edu

[†]These authors contributed equally to this work

**Competing interests:** The authors declare that no competing interests exist.

*Xiong et al., 2010*; *Knoch et al., 2014*; *Miao et al., 2017*). Most models, however, study cell polarity in the context of biochemical signaling and do not consider cell movement or deformations originated from cell mechanics. This may be relevant for nonmotile cells including yeast (*Park and Bi, 2007*; *Slaughter et al., 2009*) but might not be appropriate for motile cells where the coupling between intracellular pathways and cell shape can be crucial in determining the mode of migration (*Camley et al., 2013*; *Camley et al., 2017*). Furthermore, most of these models only focus on one specific migration mode and do not address transitions between them. Therefore, it remains an open question how cell mechanics, coupled to a biochemical signaling module, can affect spontaneous cell polarity and can determine transitions between cell migration modes.

Here we propose a novel model that couples an oscillatory biochemical module to cell mechanics. Our choice of the biochemical model was motivated by recent findings that the self-organized phosphatidylinositol (PtdIns) phosphate waves on the membrane of Dictyostelium cells exhibit characteristics of a relaxation oscillator (*Arai et al., 2010*). Our model is able to generate amoeboid-like, keratocyte-like, and oscillatory motion by varying a single mechanical parameter, the protrusive strength, without altering the biochemical signaling pathway. We determine how the transitions depend on these parameters and we show that keratocyte-like motion is driven by an emergent traveling wave whose stability is determined by the mechanical properties of the cell. Finally, we experimentally obtain all three migration modes in wild-type *Dictyostelium discoideum* and explicitly verify model predictions by reducing the actin protrusive force using the drug latrunculin B. Our model provides a unified framework to understand the relationship between cell polarity, motility and morphology determined by cellular signaling and mechanics.

## Models and results

### Model

Our two-dimensional model is composed of two modules: a biochemical module describing the dynamics of an activator-inhibitor system which works in the relaxation oscillation regime, and a mechanical module that describes the forces responsible for cell motion and shape changes (*Figure 1a*). Our biochemical module consists of a reaction-diffusion system with an activator $A$ (which can be thought of as PtdIns phosphates and thus upstream from newly-polymerized actin; *Gerhardt et al., 2014*; *Miao et al., 2019*) and an inhibitor $R$ (which can be thought of as the phosphatase PTEN). This activator and inhibitor diffuse in the cell and obey equations that reproduce the characteristic relaxation oscillation dynamics in the PtdIns lipid system (*Arai et al., 2010*; *Matsuoka and Ueda, 2018*; *Fukushima et al., 2018*):

$$\frac{\partial A}{\partial t} = D_A \nabla \cdot (\nabla A) + F(A) - G(R)A + \zeta_1(t),$$ (1)

$$\frac{\partial R}{\partial t} = D_R \nabla \cdot (\nabla R) + \frac{c_2 A - c_1 R}{\tau} + \zeta_2(t),$$ (2)

where $D_A$ and $D_R$ are the diffusion coefficients for $A$ and $R$, respectively. In these expressions, $F(A)$ is the self-activation of the activator with a functional form that is similar to previous studies: $F(A) = [k_a A^2 / (K_a^2 + A^2) + b](A_t - A)$ (*Miao et al., 2017*). The activator is inhibited by $R$ through the negative feedback $G(R) = d_1 + d_2 R$ while $R$ is linearly activated by $A$ (see Materials and methods). The timescale of the inhibitor $\tau$ is taken to be much larger than the timescale of the activator, set by $k_a$. Finally, to ensure robustness to stochasticity, we add uniformly distributed spatial white noise terms $\langle \zeta_i(\mathbf{r}, t) \zeta_j(\mathbf{r}', t') \rangle = \sigma \delta_{ij} \delta(\mathbf{r} - \mathbf{r}') \delta(t - t')$.

Nullclines for this system are shown in *Figure 1b*, where we have chosen parameters such that the fixed point is unstable and the system operates in the oscillatory regime. As a result of the separation of timescales for $A$ and $R$, the dynamics of $A$ and $R$ are characteristic of a relaxation oscillator (inset of *Figure 1b*): $A$ reaches its maximum quickly, followed by a slower relaxation phase during which the system completes the entire oscillation period.

To generate cell motion, we couple the output of the biochemical model to a mechanical module which incorporates membrane tension and protrusive forces that are proportional to the levels of activator $A$ and normal to the membrane, similar to previous studies (*Shao et al., 2010*; *Shao et al.,*

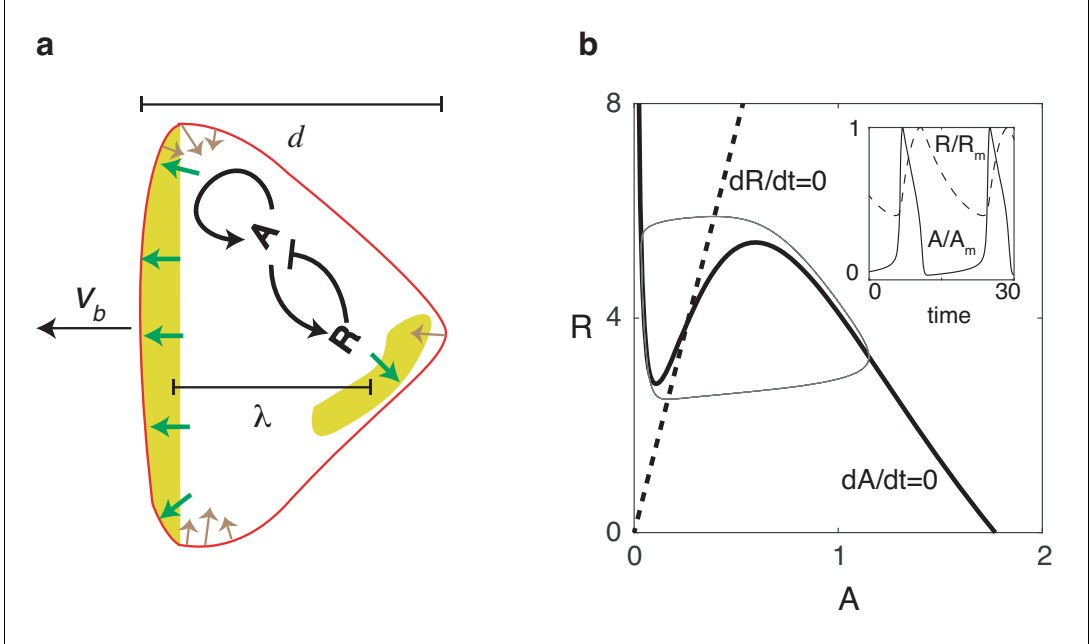

**Figure 1.** Reaction diffusion model coupled to a mechanical model. (**a**) Schematic illustration of the two-dimensional model: a self-activating activator field $A$, indicated in color, drives the movement of the cell membrane through protrusive forces that are normal to the membrane (green arrows). The membrane tension (denoted by brown arrows) is proportional to the local curvature while the cell also experiences a drag force that is proportional to the speed. One successive wave is generated behind the original one after a distance of $\lambda$. The cell's front-back distance is $d$, and the cell boundary is pushed outward with speed $v_b$. (**b**) Nullclines of the activator (solid line) and inhibitor (dashed line), along with the resulting trajectory in phase space (gray thin line). The inset shows the oscillations of $A$ and $R$, normalized by their maximum values $A_m$ and $R_m$.

DOI: https://doi.org/10.7554/eLife.48478.002

*2012*) (see Materials and methods and *Figure 1a*). To accurately capture the deformation of the cell in simulations, we use the phase field method (*Shao et al., 2010*; *Ziebert et al., 2012*; *Shao et al., 2012*; *Najem and Grant, 2013*; *Marth and Voigt, 2014*; *Camley et al., 2017*; *Cao et al., 2019*). Here, an auxiliary field $\phi$ is introduced to distinguish between the cell interior ($\phi = 1$) and exterior ($\phi = 0$), and the membrane can be efficiently tracked by the contour $\phi = 1/2$. Coupling this field to the reaction-diffusion equations can guarantee that no-flux boundary conditions at the membrane are automatically satisfied (*Kockelkoren et al., 2003*). The evolution of the phase-field is then determined by the force balance equation:

$$\xi \frac{\partial \phi}{\partial t} = \eta M(A) |\nabla \phi| - \frac{\delta H(\phi)}{\epsilon \delta \phi}, \tag{3}$$

where $\xi$ is a friction coefficient, $\epsilon$ is the boundary width of the phase field, and $H(\phi)$ is a Hamiltonian energy including the membrane tension, parameterized by $\gamma$ and area conservation (see Materials and methods). The first term on the right hand side describes the actin protrusive force, parameterized by $\eta$, and acts on the cell boundary since $|\nabla \phi|$ is non-zero only in a region with width $\epsilon$ formulates the dependence of the protrusive force on the activator levels and is taken to be sigmoidal: $M(A) = A^n/(A^n + A_0^n)$, where $n$ is a Hill coefficient. As initial conditions, we use a disk with radius $r$ with area $S = \pi r^2$ and set $A = R = 0$. Default parameter values for our model are estimated from experimental data and given in *Table 1*.

Cell motion is quantified by computing the velocity of the center of mass of the cell. Furthermore, we record the trajectory of the center of mass and compute its average curvature $\langle \kappa \rangle = \int \kappa(l) dl / L$, where $k(l)$ is the local curvature, and $L$ is the total length of the trajectory. These quantities can be used to distinguish between different migration modes (see Results and Materials and methods).

**Table 1.** Model Parameters.

| Parameter | Description | Value |
|---|---|---|
| $\gamma$ | Tension | 2 pN μm |
| $\epsilon$ | Width of phase field | 2 μm |
| $B_S$ | Cell area conservation strength | 10 pN/μm$^2$ |
| $\xi$ | Friction coefficient | 10 pN s/μm$^2$ |
| $n$ | Hill coefficient of protrusive force | 3 |
| $k_a$ | Activation rate | 10 s$^{-1}$ |
| $K_a$ | Activation threshold | 1 μM |
| $b$ | Basal activation rate | 0.1 s$^{-1}$ |
| $A_t$ | Total activator concentration | 2 μM |
| $d_1$ | Basal degradation rate | 1 s$^{-1}$ |
| $d_2$ | Degradation rate from inhibitor | 1 μM$^{-1}$s$^{-1}$ |
| $c_1$ | Inhibitor degradation coeffecient | 1 |
| $c_2$ | Inhibitor activation coefficient | 15 |
| $\tau$ | Time scale of negative feedback | 10 s |
| $D_A$ | Activator diffusion coefficient | 0.5 μm$^2$/s |
| $D_R$ | Inhibitor diffusion coefficient | 0.5 μm$^2$/s |
| $\sigma$ | Noise intensity | 0.01 μM$^2$/μm$^2$/s |
| $\Delta t$ | Time step | 0.001 s |
| $n, m$ | Space grid size | 256,256 |
| $L_{x,y}$ | Space size | 50,50 μm |

DOI: https://doi.org/10.7554/eLife.48478.003

## Computational results

We first examine the possible migration modes as a function of the protrusive strength $\eta$ for fixed area $S$, parametrized by the radius $r$ of the disk used as initial condition, and default parameters. As shown in *Figure 2*, there are three distinct cell migration modes. When $\eta$ is small, activator waves initiate in the interior and propagate to the cell boundary. However, the protrusive force is too small to cause significant membrane displacement, as also can be seen from the trajectory in *Figure 2b*. Consequently, the cell is almost non-motile and the activator and inhibitor field show oscillatory behavior (*Figure 2a* I and b and *Video 1*).

As $\eta$ increases, an activator wave that reaches the boundary can create membrane deformations, leading to the breaking of spatial homogeneity. This wave, however, is competing with other traveling waves that emerge from random positions. Consequently, the cell exhibits transient polarity, moves in constantly changing directions, and displays amoeboid-like migration (*Figure 2a* II and b and *Video 2*).

When $\eta$ is increased further, protrusions generated by activator waves reaching the cell boundary become even larger. As a result of the coupling between the waves and membrane mechanics, a single traveling wave will emerge within the cell, characterized by a broad and stationary band of high levels of activator. This wave pushes the membrane forward in a persistent direction with constant speed and the cell will adopt a steady keratocyte-like morphology, even in the presence of noise (*Figure 2a* III and b and *Video 3*).

The transition from oscillatory dynamics to amoeboid-like unstable cell motion can be understood by considering the coupling between the traveling waves and membrane motion. In Materials and methods we show that these traveling waves, that emerge naturally in systems of relaxation oscillators (*Kopell and Howard, 1973*; *Keener, 1980*; *Murray, 2002*) are stable as long as the activator front can 'outrun' the inhibitor's spreading speed. This condition results in a minimal wave speed $c_{\min}$ that depends on $D_R$ and $\tau$ (see Materials and methods). The activator wave pushes the membrane outward and will keep propagating as long as the boundary can keep up with the

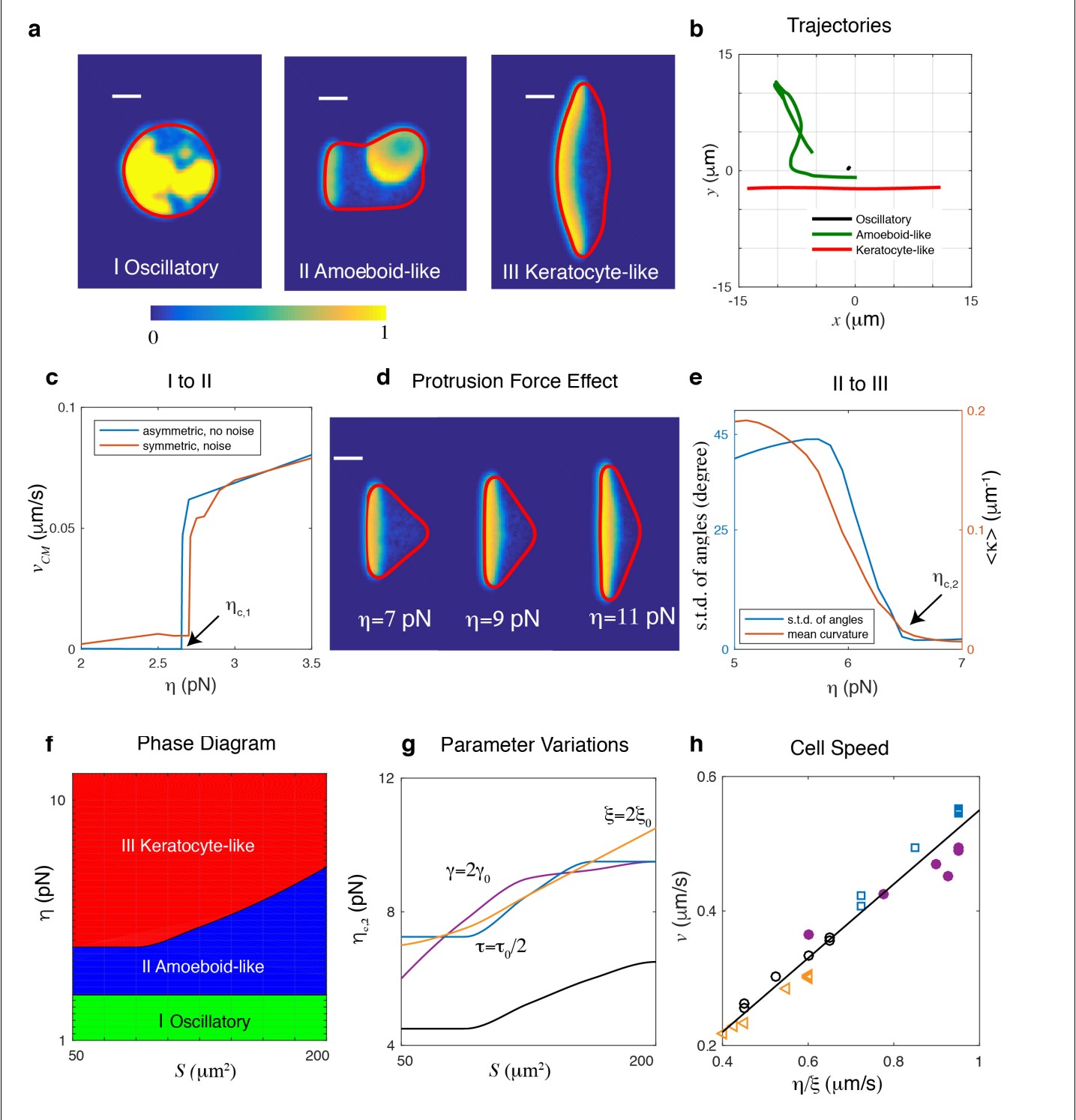

**Figure 2.** Different cell migration modes can be captured in the model by varying the protrusive strength $\eta$. (a) Snapshots of a simulated cell showing (I) an oscillatory cell ($\eta = 2\,\text{pN}$), (II) an amoeboid-like cell ($\eta = 4\,\text{pN}$), and (III) a keratocyte-like cell ($\eta = 11\,\text{pN}$). All other parameters were assigned the default values and r = 8µm. Here, the activator concentration is shown using the color scale and the cell membrane is plotted as a red line (scale bar µm). (b) The trajectories of the three cells in (a). (c) The transition from oscillatory cell to amoeboid-like cell, with speed of the center of mass of a cell as a function of protrusion strength $\eta$ for r = 8µm. The red curve represents results from initial conditions where noise is added to a homogeneous $A$ and $R$ field while the blue curve corresponds to simulations in which the initial activator is asymmetric. Cells become non-motile at a critical value of protrusion strength, $\eta_{c,1}$. (d) Increasing the protrusive force $\eta$ will result in flatter fronts in keratocyte-like cells and a decreased front-back distance. The

*Figure 2 continued on next page*

*Figure 2 continued*

simulations are carried out for fixed cell area $S$. (e) The transition from amoeboid-like cell to keratocyte-like cell quantified by either the average curvature along a trajectory or the standard deviation of the angles of trajectory points as a function of protrusion strength $\eta$ (r = 8μm). Cell moves unidirectionally when the protrusion strength $\eta > \eta_{c,2}$. (f) Phase diagram determined by systematically varying $\eta$ and the initial radius of the cell, $r$. Due to strong area conservation, cell area is determined through $S = \pi r^2$. (g) The transition line of amoeboid-like cell to keratocyte-like cell for different parameter values. (h) The speed of the keratocyte-like cell as a function of $\eta/\xi$. The black line is the predicted cell speed with $v_b = \alpha \eta / \xi$, where $\alpha \approx 0.55$. Symbols represent simulations using different parameter variations: empty circles, default parameters; triangles, $\xi = 2\xi_0$; filled circles, $\gamma = 2\gamma_0$; squares, $\tau = \tau_0/2$.

DOI: https://doi.org/10.7554/eLife.48478.004

The following figure supplements are available for figure 2:

**Figure supplement 1.** Speed of keratocyte-like cells as a function of the surface tension and timescale of the inhibitor.
DOI: https://doi.org/10.7554/eLife.48478.005
**Figure supplement 2.** Critical protrusion strength $\eta_{c,1}$ as a function of the inhibitor's diffusion constant and time scale.
DOI: https://doi.org/10.7554/eLife.48478.006
**Figure supplement 3.** Effects of tension.
DOI: https://doi.org/10.7554/eLife.48478.007
**Figure supplement 4.** Parameter variations in the model.
DOI: https://doi.org/10.7554/eLife.48478.008
**Figure supplement 5.** Oscillatory cells for strong and weak area conservation.
DOI: https://doi.org/10.7554/eLife.48478.009
**Figure supplement 6.** Excitable dynamics can reproduce identical qualitative results.
DOI: https://doi.org/10.7554/eLife.48478.010

wave speed. In our model, the membrane is pushed outward by a protrusive force resulting in a speed approximately given by $v_b \sim \alpha \eta / \xi$, where $\alpha$ is the boundary-averaged value of $M(A)$, that depends on mechanical parameters and is independent of biochemical parameters (see Materials and methods and *Figure 2—figure supplement 1*). This implies that if the speed of the membrane is less that the minimum speed of the activator wave ($v_b < c_{min}$) there will be no significant membrane motion. On the other hand, when $v_b > c_{min}$, a traveling wave can be selected by matching the wave speed and the boundary speed:

$$c = v_b \sim \frac{\alpha \eta}{\xi}. \qquad (4)$$

The above equation indicates that when $\eta > \xi c_{min}/\alpha$, the cell will break its symmetric shape through traveling waves that deform the membrane.

In simulations, the critical value of $\eta$, $\eta_{c,1}$, for which the oscillating cell becomes amoeboid-like can be determined by slowly increasing $\eta$ and computing the center-of-mass speed of the cell, $v_{CM}$. Oscillatory cells are then defined as cells with a vanishing center of mass speed (see also Materials and methods). The transition between oscillatory and amoeboid-like cells is shown in *Figure 2c* where we plot $v_{CM}$ as a function of protrusion strength $\eta$ for cells with as initial condition either homogeneous solutions in $A$ and $R$ that are perturbed with noise (blue curve) or asymmetric distributions of $A$ and $R$ (red curve). For both initial conditions, the speed shows a subcritical bifurcation at a critical value of $\eta_{c,1}$, above which a non-zero cell speed emerges. Furthermore, these simulations reveal that, as argued above, $\eta_{c,1}$ depends on both $D_R$ and $\tau$ (*Figure 2—figure supplement 2*).

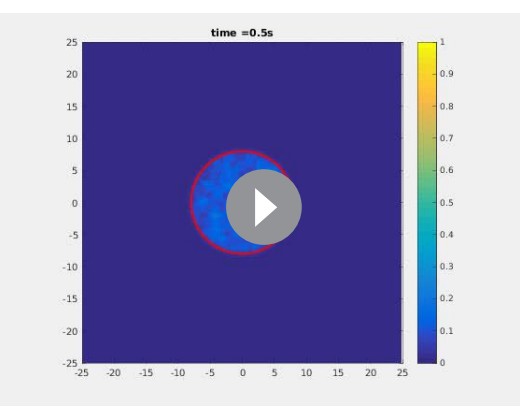

**Video 1.** Simulation results for r = 8μm and $\eta = 2\,$pN. Here, and in all other simulation videos, the activator concentration is shown using a color scale (see Figures).
DOI: https://doi.org/10.7554/eLife.48478.011

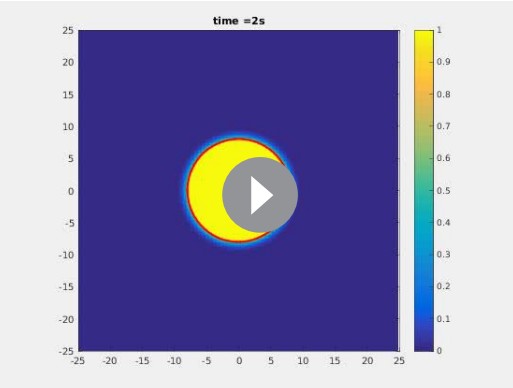

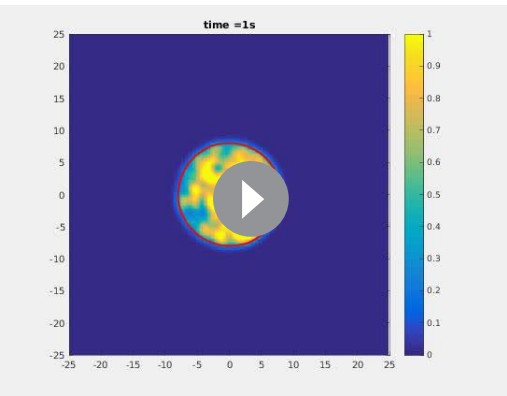

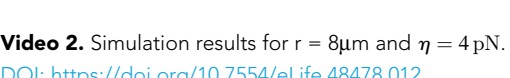

**Video 2.** Simulation results for r = 8μm and $\eta = 4\,\mathrm{pN}$. DOI: https://doi.org/10.7554/eLife.48478.012

**Video 3.** Simulation results for r = 8μm and $\eta = 11\,\mathrm{pN}$. DOI: https://doi.org/10.7554/eLife.48478.013

Once a traveling wave is able to generate membrane deformations, why does it not always result in stable, keratocyte-like motion with a single traveling wave in the cell's interior? Notice that in our model, if the spatial extent between the cell front and the back, $d$, is larger than the wavelength of the activator wave $\lambda$ (schematically shown in *Figure 1a*), a new wave will be generated behind the original one. This wavelength can be approximated by $\lambda \approx \sqrt{2D\tau}$ (where we have taken $D_A = D_R = D$ for simplicity) such that stable keratocyte-like cells driven by a single wave are only possible when $d<\lambda$. This front-back distance, however, depends on the balance between protrusive force and membrane tension at the traveling wave's lateral ends. Increasing values of $\eta$ result in a broader front and therefore smaller values of $d$ (*Figure 2d*). As a consequence, as $\eta$ is increased, waves propagating within the cell eventually become stable, resulting in a single, propagating wave and a cell with a keratocyte-like morphology. For a discussion on the role of tension we refer to Materials and methods and *Figure 2—figure supplement 3*.

To distinguish between keratocyte-like and amoeboid-like cells we compute the average curvature of the center of mass trajectory $\langle\kappa\rangle$ (see Materials and methods). For keratocyte-like cells, which move in a more persistent way, $\langle\kappa\rangle$ will take on small values while for amoeboid-like cells $\langle\kappa\rangle$ will become large. This is shown in *Figure 2e* where we plot $\langle\kappa\rangle$ as a function of protrusion strength (red curve). The transition from unstable to stable polarity, and therefore keratocyte-like motion, happens smoothly in a narrow region of $\eta$ and the critical value $\eta_{c,2}$ can be defined as the point for which $\langle\kappa\rangle = 0.02\mu m^{-1}$. Alternatively, and as in the experiments, we can compute the standard deviation of angles of trajectory points, taken at fixed intervals (see Materials and methods). As illustrated by the blue curve in *Figure 2e*, this standard deviation also decreases rapidly as $\eta$ is increased and thus provides an alternative method to distinguish the two cell migration modes.

Our analysis also implies that for equal protrusive strengths larger cells will be less stable. These cells have a larger area which allows for the nucleation of a new wave front which destabilizes the cell. For equal size cells, those with a smaller protrusion strength would have a larger font-back distance (*Figure 2d*) and therefore more space to generate a new wave, potentially destabilizing them. Consequently, decreasing the protrusion would destabilize larger keratocyte-like cells and only smaller keratocyte-like cells will remain.

From the above analysis, it becomes clear that the protrusive strength and size of initial disk, and thus cell area, are critical parameters in determining the stability of the polarity established by interactions between traveling waves and moving boundaries. In simulations, we therefore determine the phase diagram in the $(\eta, S)$ space by systematically varying the initial radius (with step size 0.5 μm) and protrusive force (with step size 0.25 pN) while keeping all other parameters fixed (*Figure 2f*). We constrain our cell area to be within the physiologically relevant range with a initial radius between r = 4μm and r = 8μm, corresponding to an area between S ~ 50μm₂ and S ~ 200μm₂ (see Materials and methods and *Figure 2—figure supplement 4* for an extension of the phase space to larger values of $r$). As stated above, there are three distinct phases, corresponding to the three different cell migration modes of *Figure 2a*. The transition from oscillatory to amoeboid-like motility

occurs at small $\eta$ and is independent of cell size, as predicted in *Equation 4*. The transition from unstable to stable polarity is quantified by $\eta_{c,2}$, which increases for increasing values of $r$ and thus $S$.

The latter transition depends on parameters that affect either $d$ or $\lambda$ (*Figure 2g*). For example, we can reduce the timescale of the inhibitor $\tau$ to half the value reported in *Table 1* ($\tau_0$) which leads to a decrease of $\lambda$ and should lead to larger values of $\eta_{c,2}$. Secondly, for increasing values of the membrane tension the transition occurs for larger values of $\eta$. This can be understood by realizing that an increase in the membrane tension will reduce the cell's deformability. Therefore, the curvature of the cell's front will decrease, which will increase the front-back distance and thus the critical protrusion strength. Finally, increasing value of the friction coefficient $\xi$ should, according to *Equation 4*, lead to a decrease in the membrane speed. Since the biochemical wave speed $c$ is unchanged, the transition from amoeboid-like to keratocyte-like motion should occur for larger values of $\eta$. All those predictions are confirmed in our simulations, as shown in *Figure 2g*. Importantly, we find the speed of the keratocyte-like cell in all the situations is linearly dependent on $\eta/\xi$, and independent of other parameters, as predicted by *Equation 4* (*Figure 2h*).

In summary, our model predicts that a sufficient decrease of $\eta$ can destabilize keratocyte-like cells, resulting in cells that employ amoeboid-like migration, and can transform keratocyte-like and unstable cells into oscillatory cells. Furthermore, for decreasing protrusive force, the keratocyte-like cells should have a reduced speed and cell size.

## Experimental results

To test our model predictions, we carry out experiments using wild-type Dictyostelium cells (see Materials and methods). In vegetative food-rich conditions, during which food is plentiful, most cells migrate randomly using amoeboid-like motion. Starvation triggers cell-cell signaling after which cells become elongated and perform chemotaxis. However, we found that starving cells for 6 hr at sufficiently low density is enough to prevent cell-cell signaling. Under these conditions, the majority of cells still moves as amoeboid-like cells but a significant fraction, approximately 20–50%, migrates in a keratocyte-like fashion (*Video 4*). These cells adopt a fan-shaped morphology and move unidirectionally, as was also observed in certain Dictyostelium mutants (*Asano et al., 2004*). Employing these low density conditions, we can alter the cell's protrusive force by interrupting actin polymerization using the drug latrunculin B, an inhibitor of actin activity. Our model predicts that as the concentration of latrunculin increases, keratocyte-like cells are more likely to switch to unstable or oscillatory cells. Furthermore, the size and speed of the remaining keratocyte-like cells should decrease.

A snapshot of starved cells is shown in *Figure 3a*, before (left panel) and after exposure to latrunculin (right panel). Higher magnification plots of the amoeboid-like cells (top two panels) and of a keratocyte-like cell are shown to the right. In these panels, the actin distribution is visualized with the fluorescent marker limE-GFP. As can be seen by comparing the snapshots in *Figure 3a*, the number of keratocyte-like cells decreases after the exposure of latrunculin. This decrease is due to keratocyte-like cells becoming unstable and switching to the amoeboid-like mode of migration. We quantify the percentage of keratocyte-like cells, as well as the speed and shape, as a function of time for different concentrations of latrunculin for at least 100 cells. As shown in *Figure 3b*, the percentage of keratocyte-like cells decreases upon the introduction of latrunculin. Furthermore, this decrease becomes more pronounced as the concentration of latrunculin is increased (inset *Figure 3b*), consistent with our model prediction.

To further verify the model predictions, we quantify the cell area $S$ and cell speed for the

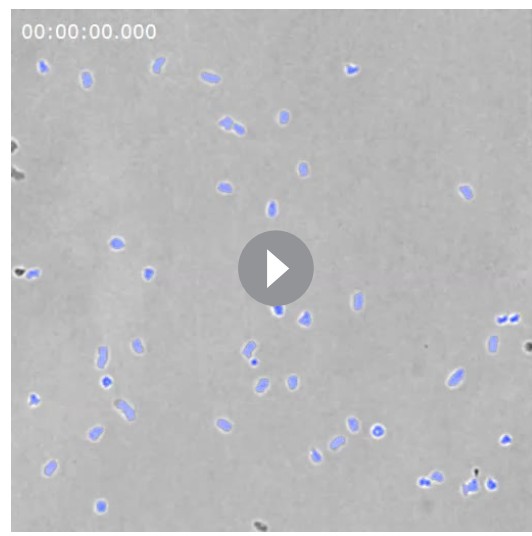

**Video 4.** Experimental results with 2μM Latrunculin B added at time 1h45min.
DOI: https://doi.org/10.7554/eLife.48478.014

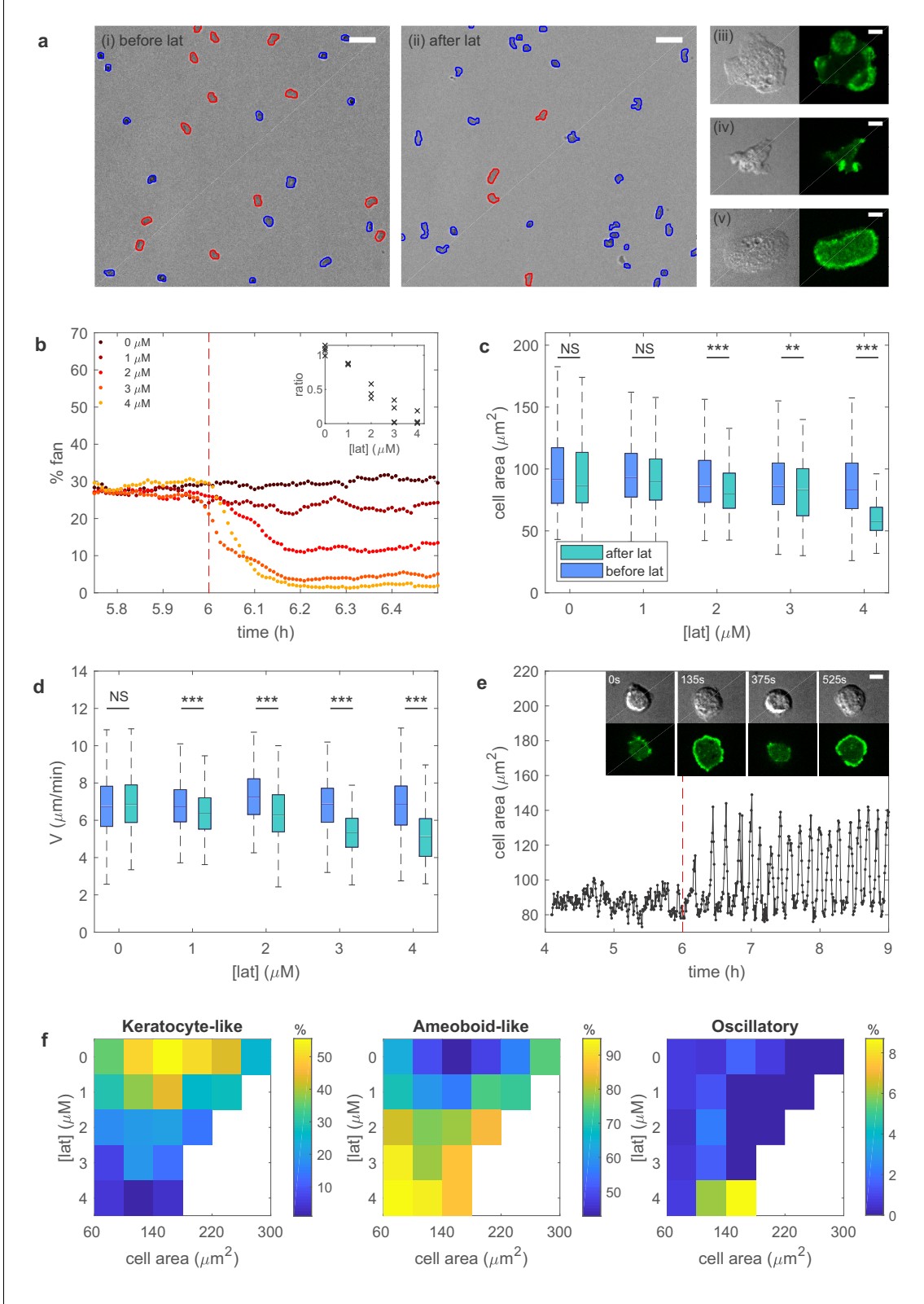

**Figure 3.** Experiments reveal different migration modes in Dictyostelium cells. (a) Snapshot of starved Dictyostelium cells before (left) and after (right) exposure to latrunculin B. Amoeboid-like cells are outlined in blue while keratocyte-like cells are outlined in red (scale bar: 50 μm). The panels on the right show high magnification views of amoeboid-like (top two) and keratocyte-like cells in which the freshly polymerized actin is visualized with limE-GFP (scale bar: 5 μm). (b) Percentage of keratocyte-like cells as a function of time for different concentrations of latrunculin B (introduced at 6 hr,

*Figure 3 continued on next page*

*Figure 3 continued*

dashed line). Inset shows the ratio of keratocyte-like to all cells as a function of the latrunculin concentration for three repeats. (**c**) The cell area size of keratocyte-like cells before and after latrunculin exposure as a function of concentration. (**d**) The speed of keratocyte-like cells before and after latrunculin exposure as a function of concentration. (**e**) Basal cell area as a function of time for a cell that transitioned from amoeboid-like to oscillatory. Insets show snapshots of the cell at different time points (scale bar: 5 μm). (**f**) Percentage of cells in the keratocyte-like, amoeboid-like, and oscillatory mode of migration in the phase space spanned by cell area and latrunculin concentration. Percentage for each mode is visualized using the color bars. The number of cells for each data point varies between 7 and >1000 and white corresponds to a data point with fewer than seven total cells.

DOI: https://doi.org/10.7554/eLife.48478.015

The following figure supplements are available for figure 3:

**Figure supplement 1.** The speed of amoeboid and keratocyte-like cells as a function of latrunculin concentration for different cell areas.

DOI: https://doi.org/10.7554/eLife.48478.016

**Figure supplement 2.** The average period and coefficient of variation for area oscillations in oscillating cells after the exposure to latrunculin.

DOI: https://doi.org/10.7554/eLife.48478.017

keratocyte-like cells. Both the area (*Figure 3c*) and speed (*Figure 3d*) decrease after the introduction of latrunculin. This decrease becomes more significant for larger concentrations of latrunculin, again consistent with our predictions. The effect of latrunculin is further shown in *Figure 3—figure supplement 1* where we plot the speed of the amoeboid and keratocyte-like cells for different areas as a function of latrunculin concentration. Speed decreases for increasing latrunculin concentration, indicating that the protrusive force is reduced in the presence of latrunculin.

Finally, our model predicts that, with a relaxed area conservation, a sufficient reduction of protrusive strength results in the appearance of oscillatory cells with oscillating basal area size (*Figure 2—figure supplement 5*). Indeed, after the exposure to latrunculin, a small fraction of cells are observed to display oscillatory behavior characterized by repeated cycle of spreading and contraction, resulting in a basal surface area that oscillates, similar to the engineered oscillatory cells of *Miao et al. (2017)* (*Figure 3e*). Cell tracking reveals that these cells originate through a transition from the unstable, amoeboid-like state to the oscillatory state. Interestingly, the observed oscillation in surface area is often very regular (*Figure 3e*). We find that the average period of different cells is largely independent of the latrunculin concentration, ranging from $6.2 \pm 0.9$ min for 1 μM to $6.6 \pm 0.9$ min for 4 μM, while the coefficient of variation, defined as the ratio of the standard deviation and the mean, for single cell periods varies between $0.31 \pm 0.12$ (1 μM) and $0.20 \pm 0.10$ (4 μM; *Figure 3—figure supplement 2*).

Our experimental results can be summarized by identifying the migration mode of cells for different values of the cell area and the latrunculin concentration, thus constructing a phase diagram that can be compared to the computational one (*Figure 2f*). Due to cell-to-cell variability, we find a distribution of migration modes for each point in this phase diagram. This is shown in *Figure 3f* where we plot, using a separate color scale for each migration mode, the percentage of keratocyte-like, amoeboid-like, and oscillatory mode of migration for different values of the cell area and latrunculin concentration. The resulting experimental phase diagrams agree well with the computational phase diagram presented in *Figure 2f*. Without latrunculin, many cells migrate using the keratocyte-like mode. Exposing cells to latrunculin, and thus reducing the protrusive force, results in a shift from keratocyte-like towards amoeboid-like cells. Furthermore, for the maximum value of the latrunculin concentration, almost all keratocyte-like cells are destabilized and a small proportion of cells ($\approx 9\%$) are in the oscillatory mode. Thus, our experimental results are in good agreement with the model predictions.

## Discussion

In this paper, we propose a simple but unified paradigm to understand cell migration and cell morphology. As in previous modeling studies (*Nishimura et al., 2009*; *Nishimura et al., 2012*; *Miao et al., 2017*; *Miao et al., 2019*), our model displays different migration modes. These modes can be induced by varying the protrusive force which is attractive since the switching of these modes can occur on a timescale that is shorter than gene expression timescales (*Figure 3b*), suggesting that a single model with conserved components should be able to capture all three modes. Importantly, our model predictions are verified in experiments using wild-type Dictyostelium cells which,

under our conditions, exhibit all three migration modes. These experiments show that upon the introduction of latrunculin the speed of keratocyte-like cells decrease. This is perhaps not surprising since latrunculin inhibits actin polymerization which can be expected to result in smaller cell speeds. Our experiments also show, however, that the area of the moving keratocyte-like cells decreases in the presence of latrunculin. In addition, and more importantly, our experiments demonstrate that transitions between the migration modes can be brought about by reducing the protrusive strength of the actin network. These non-trivial effects of latrunculin are consistent with the predictions of our model and are also captured in our experimental phase diagrams (*Figure 3f*).

Key in our model is the coupling of traveling waves generated through biochemical signaling and cell mechanics. Our main finding is that cell migration is driven by the traveling waves and that persistent propagation of these waves result in keratocyte-like cells with a broad and stable front. This stable front is only present if the front-back distance is smaller than the biochemical wavelength. Reducing the protrusive strength results in a larger front-back distance, resulting in unstable, amoeboid-like migration. For even smaller values of the protrusive strength, cells display oscillatory behavior. For these values, the membrane speed is smaller than the minimum biochemical wave speed.

Several recent computational studies have addressed switching between different migration modes. For example, Nishimura et al. have presented a model that includes actin and cortical factors, control factors of actin polymerization, and have shown that feedback between cell shape deformations and the spatially distributed control factors can result in amoeboid-like motion (*Nishimura et al., 2009*; *Nishimura et al., 2012*) . Furthermore, changing the rate of polymerization as well as the threshold of polymerization in the model can result in transitions between amoeboid-like and keratocyte-like cells (*Nishimura et al., 2009*; *Nishimura et al., 2012*). In addition, Miao et al. have proposed a model that can generate all three migration modes observed in experiments of engineered Dictyostelium cells (*Miao et al., 2017*; *Miao et al., 2019*). This model contains an excitable network and the different migration modes can be generated by altering the threshold of this network. Our current model is distinct from these studies in several ways. First, the migration mode transitions in our model are induced by the mechanical module with the same biochemical components, while in other models the transitions are generated by changing the dynamics of the biochemical signaling pathways from, for example excitable to oscillatory (*Miao et al., 2017*), or the threshold of actin polymerization (*Nishimura et al., 2009*; *Nishimura et al., 2012*). In addition, the biochemical module in our model is much simpler and only contains an activator and inhibitor while the model of *Miao et al. (2017)* requires additional feedback from a postulated polarity module. Of course, our work does not exclude the existence of this polarity module but it shows that we can explain the observed cell morphologies and movement within a minimal framework of coupling two biochemical components and cell mechanics. Second, based on the earlier measurements of *Arai et al. (2010)*, our model assumes that the biochemical module operates as a relaxation oscillator. As a result, and in contrast to *Nishimura et al. (2009)* and *Nishimura et al. (2012)*, our model is able to generate oscillatory cells. This is also in contrast to the model of *Miao et al. (2017)*, which uses nested excitable networks. Note however, that in our model we can tune the negative feedback to make the biochemical module operate in the excitable regime. We have explicitly verified that qualitatively similar migration modes and transitions are observed if our model is excitable (*Figure 2—figure supplement 6*). In the excitable version of our model, however, the oscillations in the non-motile mode are, in general, less regular and periodic than the ones obtained in the relaxation oscillator version. From the statistical features of the periods obtained from oscillatory cells in experiments, it is likely that the cellular signaling dynamics can be most accurately described by relaxation oscillation models. As a final distinction, we point out that the biochemical and mechanical module in the model of *Miao et al. (2017)* are solved separately on a 1D ring while in *Nishimura et al. (2009)* and *Nishimura et al. (2012)*, the biochemical reactions are coupled to cell deformation in a lattice model. As a result, the keratocyte-like cells in *Miao et al. (2017)* display constant excitations at the front that travel along the membrane in the lateral direction rather than stationary activator bands, as observed in the experiments and in our model (*Figure 3a(v)*). Furthermore, the keratotyte-like cells in *Nishimura et al. (2009)* and *Nishimura et al. (2012)* have similar shape to the ones generated in our model but have less persistent directionality.

Several future extensions of our study are possible. First, our current study is restricted to two dimensional geometries while actual cell motion is of course three dimensional. Extending our model to 3D would allow us to relax the area conservation constraint and should result in cells for which the

basal surface area show clear oscillations that are coupled to extensions away from the surface (*Figure 2—figure supplement 5*). Second, we have ignored fluid flow within the cytosol, which may play a role in signaling and polarity formation. Including fluid flow, which adds considerable computational complexity to the model (*Shao et al., 2012*), will be part of future extensions. Third, it should be possible to couple the biochemical model to upstream chemotaxis pathways, allowing it to address directed motion or more complex pathways. In addition, it should be possible to consider multiple parallel and excitable pathways which may regulate cell motility in chemotaxis (*Tanabe et al., 2018*) and models with more molecular details (*Matsuoka and Ueda, 2018*; *Fukushima et al., 2018*). Fourth, it would be interesting to compare wave dynamics obtained in our model with waves observed in giant Dictyostelium cells (*Gerhardt et al., 2014*). In addition, alternative biochemical models in which parameters determine the qualitatively different dynamics can be studied (*Miao et al., 2017*). Furthermore, our study predictions may also be verified in other cell types. For example, we predict that over-expression of actin in fast moving cells should result in cells migrating with keratocyte-like morphologies while disturbing actin polymerization in keratocytes could lead to unstable migration. Finally, it would be interesting to determine how the feedback between mechanical and biochemical modules can potentially help understand other cell migration processes.

## Materials and methods

### Full model

Our model for the cell boundary and cell motion is detailed in earlier studies (*Shao et al., 2010*; *Shao et al., 2012*; *Camley et al., 2014*). Briefly, we model the cell boundary as an interface with tension, driven in this study by activator $A$ at the front. Cell motion obeys the overdamped force balance equation $\boldsymbol{F}_{act} + \boldsymbol{F}_{mem} + \boldsymbol{F}_{area} + \boldsymbol{F}_{fric} = 0$ where $\boldsymbol{F}_{act}$ is the active force proportional to the activator concentration; $\boldsymbol{F}_{mem}$ describes the membrane tension (line tension since we are modeling a 2D cell); $\boldsymbol{F}_{area}$ represents area conservation to prevent cells from expanding or shrinking indefinitely and $\boldsymbol{F}_{fric}$ is a friction force. The active force from the activator is governed by $\boldsymbol{F}_{act} = \eta M(A)\hat{\boldsymbol{n}}$, where $\hat{\boldsymbol{n}} = -\nabla\phi/|\nabla\phi|$ is the outward-pointing normal direction of the membrane, and $M(A) = A^3/(A^3 + A_0^3)$ where $A_0$ represents a threshold value for activation of protrusive force. The membrane tension force is computed using the functional derivative (*Camley et al., 2014*) $\boldsymbol{F}_{mem} = \frac{\delta H_{tension}(\phi)}{\delta\phi}\nabla\phi/\delta_\epsilon$, with $\delta_\epsilon = \epsilon|\nabla\phi|^2$ and

$$H_{tension}(\phi) = \gamma \int (\frac{\epsilon}{2}|\nabla\phi|^2 + \frac{G(\phi)}{\epsilon})d^2\boldsymbol{r}. \tag{5}$$

Here, $G(\phi)$ is a double well potential with minima at $\phi = 1$ and $\phi = 0$. As in our earlier work (*Camley et al., 2017*) we neglect membrane bending and we have verified that it does not qualitatively change the results. We implement area conservation as $\boldsymbol{F}_{area} = B_S(\int \phi d^2\boldsymbol{r} - S_0)\hat{\boldsymbol{n}}$ where $B_S$ represents the strength of the area conservation and $S_0$ is the prescribed area size determined by initial cell radius $r$ through $S_0 = \pi r^2$. The friction is $\boldsymbol{F}_{fric} = \xi\boldsymbol{v}$ so that $\boldsymbol{v}$ is obtained from the force balance equation: $\boldsymbol{v} = (\boldsymbol{F}_{act} + \boldsymbol{F}_{mem} + \boldsymbol{F}_{area})/\xi$. Note that the friction coefficient takes into account the interaction with the substrate, and that fluid drag can be ignored (*Del Alamo et al., 2007*). The motion of the phase field $\phi$ is then determined by the advective equation $\partial\phi/\partial t = -\boldsymbol{v}\cdot\nabla\phi$. Finally, coupling the phase field equations to the reaction-diffusion equations presented in the main text, we arrive at the full equations:

$$\frac{\partial(\phi A)}{\partial t} = D_A\nabla\cdot(\phi\nabla A) + \phi[(\frac{k_a A^2}{K_a^2 + A^2} + b)(A_t - A) - (d_1 + d_2 R)A + \zeta_1(t)], \tag{6}$$

$$\frac{\partial(\phi R)}{\partial t} = D_R\nabla\cdot(\phi\nabla R) + \phi[\frac{c_2 A - c_1 R}{\tau} + \zeta_2(t)], \tag{7}$$

$$\xi\frac{\partial\phi}{\partial t} = \eta M(A)|\nabla\phi| + \gamma(\nabla^2\phi - \frac{G'(\phi)}{\epsilon^2}) - B_S(\int \phi d^2\boldsymbol{r} - S_0)|\nabla\phi|. \tag{8}$$

Through the coupling of $\phi$ to the reaction-diffusion equations, all reaction and diffusion processes are constrained to be inside the auxiliary field (*Kockelkoren et al., 2003*). The membrane tension parameter is similar to the one used in earlier studies (*Shao et al., 2010*; *Shao et al., 2012*; *Camley et al., 2014*) and taken from *Simson et al. (1998)*. The parameters of the biochemical module are estimated from experiments (*Arai et al., 2010*; *Gerhardt et al., 2014*), such that the minimum wave speed in simulations is approximately 0.12µm/s and the wavelength is about 15µm. Note, however, that we can rescale time constants to make simulations more efficient. Therefore, these values are obtained after increasing τ and time by a factor of 2 and 5, respectively.

## Numerical details

The parameters used for numerical simulations are listed in *Table 1*. Equations are evolved in a region with size of $L_x \times L_y$ = 50 × 50 µm with discrete grids of n×m = 256×256 and periodic boundary conditions are used. *Equation 8* is discretized using the forward Euler method with $\partial_t \phi = (\phi^{(n+1)} - \phi^{(n)})/\Delta t$. Derivatives are calculated using finite difference formulas: $\partial_x \phi = (\phi_{i+1,j} - \phi_{i-1,j})/(2\Delta x)$ and $\partial_x^2 \phi = (\phi_{i+1,j} + \phi_{i-1,j} - 2\phi_{i,j})/\Delta x^2$, with similar equations for the derivatives in the $y$-direction. *Equations 6 and 7* are discretized using the forward Euler scheme with $\partial_t(\phi A) = \phi^{(n)}(A^{(n+1)} - A^{(n)})/\Delta t + A^{(n)}(\phi^{(n+1)} - \phi^{(n)})/\Delta t$. The diffusion terms $\nabla \cdot (\phi \nabla A)$ are also approximated using finite difference. The x-term, for example, reads $[(\phi_{i+1,j} + \phi_{i,j})(A_{i+1,j} - A_{i,j})/(2\Delta x) - (\phi_{i,j} + \phi_{i-1,j})(A_{i,j} - A_{i-1,j})/(2\Delta x)]/\Delta x$. The white noise terms are simulated as Wiener processes with $\zeta(t)\Delta t = \sqrt{\sigma \Delta t} N(0,1)$. As initial condition for $\phi$, we use a disk $\phi = \frac{1}{2}[1 + \tanh(3(r - r_0)/\epsilon)]$, where $r_0$ is the prescribed radius and $A = R = 0$. The activator and inhibitor concentration outside the boundary is 0. To implement this boundary condition, we solve *Equations 6 and 7* only in region $\epsilon_0$ away from $\phi = 1/2$ which is $\phi > \chi = 1/2 + 1/2 \tanh(-3\epsilon_0/\epsilon) \approx 0.0025$, and leave $A = R = 0$ outside this region. Here, we have taken $\epsilon_0 = 2\mu m$.

Equations are parallelized with CUDA and simulated using GPUs. Typical simulations speeds on a high-end graphics board are less than one minute for 100 s of model time.

## Identification of migration modes

In our simulations, we track the motion of the center of mass of the cell which results in a cell trajectory. Oscillatory cells are defined as cells with a vanishing center of mass speed (see also *Figure 2c*). To distinguish amoeboid-like from keratocyte-like cells we can use one of two strategies. The first one is identical to the one used in the experiments and uses the position of the center of mass at discrete time intervals. We then compute the angle between the vector connecting consecutive points and an arbitrary fixed axis and compute the standard deviation of this distribution. Keratocyte-like cells, which move more persistently and thus have straighter trajectories, will have a smaller standard deviation than amoeboid-like cells (*Figure 2b*). The result is shown in *Figure 2e* where we plot the standard deviation for 100 consecutive intervals, separated by 3 s as a function of protrusion strength (blue curve). Alternatively, taking advantage of the high temporal resolution of the computational tracks, we can use the curvature of the cell trajectory to identify the transition between amoeboid-like and keratocyte-like cells. This is also shown in *Figure 2e* where we plot the average curvature $\langle \kappa \rangle = \int \kappa(l) dl / L$ (red curve) as a function of protrusion strength. Here, $\kappa(l)$ is the local curvature, and $L$ is the total length of the trajectory. We can then define keratocyte-like cells as those cells with a standard deviation smaller than 2° or with an average curvature smaller than 0.02 µm$^{-1}$.

To identify the migration mode in the experiments we also track the centroid of the cell. However, since the temporal resolution in the experiments are much lower than that of our simulations, we cannot employ the average curvature identification described above. Instead, we use the angle method and compute angles between two successive positions, separated by 30 s. The standard deviation of this angle is then computed for five consecutive pairs and keratocyte-like cells are defined by having a standard deviation less than 25°. To determine whether cells can be considered oscillatory we compute the cell area and evaluate three criteria. First, we compute the coefficient of variation COV (ratio of the standard deviation and the mean) of the area oscillations (computed using five consecutive frames separated by 30 s). Second, we determine the maximum (peaks) and minimum values (valleys) in a time trace of the cell area (see *Figure 3e*) and quantify the ratio of the difference between the average peak and valley and the average peak: P=(<peak>-<valley>)/

<peak>. Third, we evaluate the total amount of time $T_{tot}$ an oscillation is present. Oscillatory cells are then defined as cells with COV>5%, P>19%, and $T_{tot} > 40$ min.

## Speed of keratocyte-like cells

The local cell boundary velocity can be approximated as $\boldsymbol{v} = \hat{\boldsymbol{n}} \eta M(A)/\xi$. Since, for keratocyte-like cells, the front is almost flat (see main text *Figure 2*), the cell speed can be approximated as $v = \alpha \eta / \xi$, where $\alpha$ is the average of $M(A)$ across the boundary: $\alpha = \int_{-\epsilon}^{\epsilon} M(A)\phi dx/(2\epsilon) \approx 0.55$. Thus, the speed of keratocyte-like cells only depends on the protrusive force and the friction but not on the tension or on the inhibitor timescale. We can verify this in simulations by changing the tension $\gamma$ and the inhibitor's timescale $\tau$ while keeping other parameters fixed. The results are shown in *Figure 2—figure supplement 1* which demonstrates that the cell speed changes little if these parameters are varied.

## Traveling waves

Our relaxation oscillator system exhibits traveling waves with a minimum speed that depends on the timescale of the inhibitor and its diffusion constant. Our model can be written as $\partial_t A = D\nabla^2 A + f(A,R)$, $\partial_t R = D\nabla^2 R + g(A,R)$, where $f(A,R)$ and $g(A,R)$ can be found from *Equations 6 and 7*. First we consider the case of $\tau \to \infty$ and $D_R = 0$ so that the inhibitor is uniformly distributed and constant: $R = R_0$. The relevant equation now is $\partial_t A = D\partial_x^2 A + f(A;R_0)$. When $R_0$ is in a proper range, there are three steady states $A = A_{1,2,3}$, with $A_1 < A_2 < A_3$ and $A = A_{1,3}$ stable and $A = A_2$ unstable. For a given $R_0$, we seek for solutions of wave form $A(z) = A(x - ct)$. The excitable version of this system has been extensively studied and for the cubic reaction term $k(u - u_1)(u_2 - u)(u - u_3)$ there is a stable traveling wave solution that connects $u$ with $u_1$ and $u_3$ and has a wave speed $c = \sqrt{kD/2}(u_1 - 2u_2 + u_3)$ (*Murray, 2002*). Likewise, our model $\partial_t A = D\partial_x^2 A + f(A;R_0)$, which has the same structure as the excitable system with a cubic reaction equation, has a stable traveling wave with speed $c \approx w(A_1 - 2A_2 + A_3)$ which depends on $R_0$ through $A_{1,2,3}$. Here, $w$ is a constant that only depends on the diffusion coefficient $D_A$ and reaction rates (cf. $\sqrt{kD/2}$ for the cubic reaction equation).

For non-zero values of $\tau$ and $D_R$, the relaxation phase, characterized by the accumulation of inhibitor, sets in behind the activator's wave front. From the above analysis, it can be deduced that a necessary condition for a stable wave front is a constant profile of the inhibitor $R = R_0$ in the width of the front. For $D_R = 0$, this can only be the case when $\tau$ is large so that the reaction rate of the inhibitor is slow compared to the rate of the activator. Thus, there needs to be a separation of timescales and the system has to obey relaxation dynamics. Furthermore, for $D_R > 0$, the activator wave front is only stable if it can outrun the inhibitor's spreading speed which is proportional to $\sqrt{D_R/\tau}$. Therefore, the traveling wave is only stable if both the reaction and diffusion of the inhibitor are slow enough. In other words, stable waves will have a speed that exceeds a minimum value which depends both on $D_R$ and $\tau$. In simulations, we can change $c_{\min}$ by changing the inhibitor's diffusion coefficient $D_R$ and the timescale $\tau$. As expected, larger $D_R$ and smaller $\tau$ leads to larger $c_{\min}$, and consequently larger $\eta_{c,1}$ (*Figure 2—figure supplement 2a and b*).

## The role of tension in morphology

Tension is important to maintain the unidirectional movement of keratocyte-like cells. In our model, at the two lateral ends of the traveling wave the morphology is determined by a balance between the protrusive force and the tension which is determined by the local curvature and the parameter $\gamma$. When the protrusive force increases, a larger membrane curvature is necessary to balance the protrusive force, resulting in a flatter front and a decreased front-back distance. If $\gamma$ is not large enough, the traveling wavefront can have a turning instability: the cell will no longer migrate along a straight path and will make a turn (*Camley et al., 2017*). An example of this instability and the resulting motion is shown in *Figure 2—figure supplement 3a* where we decrease the surface tension by a factor of 2 (from $\gamma = 2$pN/μm to $\gamma = 1$pN/μm). Similar to *Camley et al. (2017)*, the cell can also be destabilized by increasing the diffusion of the activator and inhibitor. An example of a simulation showing this can be found in *Figure 2—figure supplement 3b*.

## Parameter variations

We have examined how the model results change when certain parameters are varied. For example, we can extend the $(\eta, r)$ phase space to larger values of $r$ as shown in *Figure 2—figure supplement 4a*. For cell sizes that are beyond ones observed in experiments, we find that the critical protrusive force $\eta_{c,2}$ saturates. For these large values of $r$, a dominant wave forms at the front of the cell and new waves that are generated at the back of the cell are not strong enough to break this dominant wave's persistency. The cell will move persistently in the direction of the dominant wave, with smaller waves repeatedly appearing at the back, as shown in the snapshot presented in *Figure 2—figure supplement 4a*. For keratocyte-like cells with a single wave, $d$ will saturate to the wavelength $\lambda \approx 13$ μm, as shown in *Figure 2—figure supplement 4b*. Finally, we have verified that changing the Hill coefficient in $M(A)$ does not change the phase diagram and thus the transitions in a qualitative fashion. This is shown in *Figure 2—figure supplement 4c* where we plot the boundaries between the different migration modes in the phase diagram for three different values of the Hill coefficient.

## Varying area conservation

Real cells are three-dimensional objects in which the changing of the basal surface area will be compensated by the morphological changes away from the substrate. Our model represents the cell as a two-dimensional object and therefore includes an area conservation term. Making the strength of this area conversation large in simulations allows us to define and sample the $(\eta, r)$ phase space (*Figure 2b*). To determine how this area conservation term affects the observed dynamics, we reduce the area conservation parameter from $S_B = 10$ to $S_B = 0.1$. As shown in *Figure 2—figure supplement 5*, all migration modes are still present. For small values of $\eta$, cells are oscillatory and, compared to large values of $S_B$, exhibit measurable oscillations in cell size (*Figure 2—figure supplement 5a–c*). Furthermore, increasing the value of the protrusive strengths results in amoeboid-like motion while even larger values of $\eta$ lead to keratocyte-like cells (*Figure 2—figure supplement 5d*).

## Excitable model

Our biochemical model is based on relaxation oscillation dynamics. However, it is straightforward to consider an excitable version of the model. For this, we take $c_2 = 30, \sigma = 0.1 \mu M^2 \mu m^2/s$ and keep all the other parameters same as listed in *Table 1*. For the excitable version of our biochemical module, we find similar migration mode transitions as found in the main text as well as a qualitatively similar phase diagram (*Figure 2—figure supplement 6a and b*). Specifically, we also find a nonmotile, amoeboid-like, and a keratocyte-like mode (*Figure 2—figure supplement 6a*). In the excitable case, however, perturbations are required to initiate waves and movement. As a consequence, the patterns of activator are more noisy than for the oscillation model and the non-motile cells do not exhibit oscillatory dynamics. Finally, the transition between nonmotile cells and amoeboid-like cells is also subcritical (*Figure 2—figure supplement 6c*).

## Experiments

Wild-type AX2 cells were transformed with the plasmid expressing limE-delta-coil-GFP. Cells were kept in exponential growth phase in a shaker at 22 °C in HL5 media with hygromycin (50 μg/mL). On the day before the experiment, cells were diluted to low concentration (1–2×10⁵ cells/mL) to stop the exponential growth. After 15 h-18h, cell concentration reached 2–5×10⁵ cells/mL and 10⁵ cells were plated in a 50 mm round chamber with glass bottom (WillCo). After 15 min, cells attached to the substrate and HL5 was replaced with 7 mL DB (5 mM Na₂HPO₄, 5 mM KH₂PO₄,200μM CaCl₂, 2 mM MgCl₂, pH6.5). Specified amount was diluted in 500μL of DB and added to the sample using a pipette at 6 hr of starvation.

Differential interference contrast (DIC) images are taken every 30 s in six fields of view across the sample using a 10x objective from 5h45 to 6h30 after beginning of starvation. Cell centroids, area, minor and major axis are tracked using Slidebook 6 (Intelligent Imaging Innovations). Statistical analysis of trajectories is performed in MATLAB (2018a; The Mathworks). Experimental data presented before latrunculin B are for cells between 5h45 to 6 hr of starvation, whereas effect of the drug is quantified on cells from 6h15 to 6h30. Speeds are measured using a time interval of 1 min. For each concentration, data are collected on three different days, resulting in N = 333 , N = 315, N = 385, N = 567, and N = 399 for 0, 1, 2, 3, and 4 nM latruculin, respectively. P values are computed with

the Wilcoxon rank sum test. Fluorescent images (488 nm excitation) are captured with a 63x oil objective using a spinning-disk confocal Zeiss Axio Observer inverted microscope equipped with a Roper Quantum 512SC cameras.

## Acknowledgements

We thank Brian A Camley for many useful discussion. This work was supported by the National Science Foundation under grant PHY-1707637 and HFSP number LT000371/2017 C.

## Additional information

### Funding

| Funder | Grant reference number | Author |
| --- | --- | --- |
| National Science Foundation | PHY-1707637 | Wouter-Jan Rappel |
| Human Frontier Science Program | LT000371/2017-C | Elisabeth Ghabache |

The funders had no role in study design, data collection and interpretation, or the decision to submit the work for publication.

### Author contributions

Yuansheng Cao, Conceptualization, Data curation, Software, Formal analysis, Validation, Investigation, Methodology, Writing—original draft, Writing—review and editing; Elisabeth Ghabache, Data curation, Formal analysis, Validation, Investigation, Methodology, Writing—original draft, Writing—review and editing; Wouter-Jan Rappel, Conceptualization, Data curation, Formal analysis, Supervision, Funding acquisition, Validation, Investigation, Methodology, Writing—original draft, Project administration, Writing—review and editing

### Author ORCIDs

Yuansheng Cao (iD) http://orcid.org/0000-0002-6857-6044
Elisabeth Ghabache (iD) https://orcid.org/0000-0001-9832-9354
Wouter-Jan Rappel (iD) https://orcid.org/0000-0003-3833-7197

### Decision letter and Author response

Decision letter https://doi.org/10.7554/eLife.48478.020
Author response https://doi.org/10.7554/eLife.48478.021

## Additional files

### Supplementary files

• Transparent reporting form DOI: https://doi.org/10.7554/eLife.48478.018

### Data availability

All data generated or analysed during this study are included in the manuscript and supporting files.

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
