## [Decision Letter]

Thank you for submitting your article "Plasticity of cell migration resulting from mechanochemical coupling" for consideration by *eLife*. Your article has been reviewed by three peer reviewers, and the evaluation has been overseen by a Reviewing Editor and a Senior Editor. The following individual involved in review of your submission has agreed to reveal their identity: Jayson Paulose (Reviewer #3).

The reviewers have discussed the reviews with one another and the Reviewing Editor has drafted this decision to help you prepare a revised submission.

The manuscript addresses the question of how cells switch between different migration modes by coupling mechanics and signalling. All three reviewers are generally positive about the paper. Yet they have certain concerns that would improve both the readability and strengthen the conclusions. We think it is instructive to present the reviewers comments to you, since they are for the most part convergent and they highlight what the reviewers appreciated about this work. Please note that we are aware that the answers to some of the comments are in the supplementary information, yet since the reviewers had a hard time finding them, it is likely that general readers would find this difficult as well.

During the post-review discussion among the reviewers, additional concrete suggestions emerged:

1) It would be helpful if the authors provided a precise definition of the amoeboid -> keratocyte transition in the model, and of the amoeboid phase in the experiments, and to move the relevant information (including Figure 2—figure supplement 2A and a similar panel for the amoeboid -> keratocyte transition in the model) to the main text and figures.

2) Producing a "migration-mode diagram" from the existing experimental data to provide a more concrete comparison of model prediction to data would be really helpful. The reviewers are not asking for time consuming new experiments.

Reviewer #1:

In this work, the authors address the question of how cells can switch between different migration modes, namely amoeboid-like, characterised by rapid motility induced by unstable protrusions, and keratocyte-like, characterised by stable morphology and persistent motion. To this aim, they propose a theoretical model which takes into account (i) a signalling module, made of an activator and an inhibitor, modelled as a reaction-diffusion system; (ii) a mechanical module, modelled by a phase-field method, describing the cell shape deformations as induced by tension, bending, coupling to the signalling module and external friction with the surrounding medium. Their analysis shows that: (i) increasing the coupling between the two modules allows the cells to switch between different modes of migration; (ii) as far as the activator front can outrun the inhibitor's spreading, cells are able to migrate; (iii) while the transition from a non-migrating oscillatory behaviour to the amoeboid-like mode does not depend on the cell size, that between the amoeboid-like and keratocyte-like does. To validate their results, they perform experiments on cells treated with different amounts of an actin-polymerisation inhibitor, which allows to control the feedback between signalling and mechanics. Their findings show that qualitatively the theoretical predictions are recovered. I think that this paper is beautiful and it represents a nice step forward in the general understanding of cell migration and mechano-chemical coupling. The manuscript is in general clear and well written, the literature well represented and the authors already addressed in the paper most of the major concerns I would have in principle. Yet, I list below the points I think would definitely improve the clarity and quality of the manuscript.

1) The authors define a reaction-diffusion system made of an activator and inhibitor, respectively called A and R, which works in a relaxation oscillation regime. I do understand that this choice is made in order to observe alternated phases for the activator and the authors discuss their model reduction in the Discussion. Yet, I wonder whether the inhibitor presence is really necessary and a more coarse-grained model could actually reproduce the same results.

2) I have a general feeling that the model results could be better discussed in the main text. Most of the points about the variation of parameters are described in the supplementary materials and the reasoning behind the main text is hard to follow. To make the manuscript clearer, I would also move some of the supplementary figures in the (already existing) main figures by condensing/rearranging the panels and giving them proper titles (e.g., "Protrusive force effect", etc.), once the theory discussion in the main text has been expanded.

3) The authors model cell mechanics with a phase field method. The cell shape is described by an auxiliary field that defines the boundary of the cell changing as a function of mechano-chemical forces. I have two main questions about this point. First, I wonder why the authors did not use a more realistic continuous active-viscous shell description. In the context of the phase field model, any viscous dissipation of the membrane is completely neglected and I wonder whether this would only shift the migration-mode switch appearance or would strongly modify the results. Also, in general, the parameters in the phase-field model are quite coarse grained compared to those actually measurable experimentally (e.g., surface tensions). Can the authors clarify this point? Second, the coupling between the mechanical and signalling modules is taken to be strongly non-linear via a sigmoidal function of the activator. The authors analyse in depth the dependency of the cell migration modes on the intensity of this coupling. I do think it would be worth to investigate also whether and how changing the "Hill coefficient" could lead to motility switches in the model, as this would give more information about the mathematical description of the properties of the mechano-chemical coupling in this context and possible cooperative effects at the biochemical level.

4) To my knowledge, there are not universally accepted values for the bending modulus but just very wide ranges of parameters (if I am wrong, I would be very happy to have the reference precisely quantifying this). Could the authors describe whether they investigated the role of this parameter (I have looked into the supplementary materials but did not find any mention about it)? In principle, since bending prevents changes in the mean curvature, I would expect a smoothening or even a disappearance of the switch between the different migration modes.

5) The authors perform experiments on cells where their protrusive force is modulated by interfering with actin polymerisation. Their results all agree qualitatively with their model, i.e., decreasing the protrusive force destabilises keratocyte-like migration and allows the switch to the amoeboid-like migration. This has also effects on other quantities as front-back distance, speed and size as qualitatively found by the model. Yet, I do not see anywhere any quantitative comparison between model and experiments. Can the authors clarify this and show an experimental "migration-mode diagram"? A quantitative comparison would be very interesting as it would require in principle a more general definition of protrusive force in the model, as interfering with actin polymerisation alters all mechanical parameters such as tension and bending.

Reviewer #2:

The manuscript by Cao et al. presented a model which incorporates a biochemical module to cell mechanics. The mathematical model produced different migration modes, such as amoeboid-like and keratocyte-like motility, depending on strengths of mechanical forces. The authors also evaluated the model by some experiments using *Dictyostelium* cells. It is a prevailing idea that different migration modes are exchangeable and arise from a common mechanism. This study revealed that a single mechanical parameter, the protrusive strength, enables the transition of different migration modes. However, the author's group has previously constructed similar models for cell boundary and motility. The biochemical model is based on a relaxation oscillator by Arai et al. (2010). Thus, this study likely combined the two existing models. Also, experimental evaluation is not conclusive. My major concerns are described below.

An important finding in this study is that the transition of migration modes depends on the protrusive strength (η) and cell size (r). But it is not clear what is the definition of each migration mode produced by the model. Previous work characterized the amoeboid- as well as keratocyte-like motility of *Dictyostelium* cells (Takagi et al., PLoS One, 2008: Asano et al., 2004). Does the model here recapitulate these features of real cell behaviors? Also, I am not sure how the activator is maintained at the front of a keratocyte-like cell even after the inhibitor is produced.

The model was evaluated by adding latrunculin to reduce the protrusive strength in Figure 3 but there are some issues to be addressed. First, the criteria for each migration mode is missing in the manuscript.

Second, the keratocyte-like cells may differ between the model and the experiment. The model predicted that F-actin asymmetrically localize at the cellular front but the keratocyte-like cell shown in Figure 3A has F-actin at the entire cortex. Also, the model should be evaluated by drawing the phase-diagram in the (η,r) space of *Dictyostelium* cells shown in Figure 3A. The protrusive strength could be quantified by the fluorescence intensity of limE-GFP.

Third, the authors claim that the model predicts decreases in cell area size and front-back-distance of keratocyte-like cells after latrunculin treatment in the third paragraph of the subsection “Experimental Results” but I do not understand why. I thought that a decrease in protrusion strength increases front-back-distance shown in Figure 2—figure supplement 3A.

Fourth, in addition to the study by Miao et al. which is deeply discussed in this manuscript, Nishimura et al. previously published mathematical analysis for the transition of different migration modes (Nishimura, Ueda and Sasai, 2009; 2012). The authors should discuss these works as compared with the current model.

Fifth, the current model still can operate in the excitable regime of the PtdIns phosphate module in Figure 2—figure supplement 6. The recent study showed that multiple pathways including PtdIns phosphate and sGC regulate random cell motility through their distinct excitable properties (Tanabe et al., 2018). Moreover, the model presented by Arai et al. was revised by Fukushima et al. and a new feature of PtdIns phosphate dynamics via the mutual inhibition between PIP3 and PTEN was reported to ensure polarity formation. These points should be discussed so that the presented model could be more realistically considered.

Reviewer #3:

The authors report combined numerical and experimental work on cell motility driven by the coupling between biochemical waves arising from a reaction-diffusion equation and the mechanics of cell shape. The main advance lies in coupling chemical oscillations of an activator-inhibitor system to a mechanical model of the cell: the activator pushes on the cell boundary to change its shape and position, which in turn modifies the spatial domain of the chemical reaction and influences the observed patterns. From simulations and analytical considerations of the waves supported by the reaction-diffusion system, three distinct spatiotemporal patterns are identified which are associated with specific motility and morphology signatures observed in migrating cells. Transitions between these phases are driven by various factors, of which the cell area and the strength of pushing are the most significant. The predictions of the model are tested in an experiment using *Dictyostelium* cells, in which a reduction of the pushing strength is observed to change the dominant migration mode and the cell morphologies in a manner consistent with the model predictions.

The work appears correct, complete, and well-presented overall except for the major issues below. As far as I can judge, the mechanism does stand apart from previous works in reducing the complexity and number of components needed to recover the range of cell behaviors observed, and in highlighting the interplay between chemical dynamics and isotropic mechanics (without additional polarization fields). This interplay could be relevant to a variety of other systems which rely on the feedback between reaction-diffusion chemical dynamics and cell mechanics, and the relative simplicity/generality of the model presented here makes it potentially applicable to these other situations. Therefore, I find that the results are significant and of broad interest. I have two major concerns about the methodology which I would like to see addressed before publication.

1) The nature of the transition from amoeboid-like to keratocyte-like behavior in the model is not adequately quantified. The transition is described as being from "unstable to stable polarity" and the procedure for classifying experimental trajectories is provided (subsection “Experiments”), but I did not see a rigorous definition of the phase in terms of quantities measured in simulations. How is η_c,2_ determined, and is there a sharp transition in some quantity akin to that seen in the cell speed for the stationary -> amoeboid transition (Figure 2—figure supplement 2A)? This information is needed to judge whether the amoeboid -> keratocyte transition is indeed a sharp one in the r-η plane (as suggested in the Abstract and in Figure 2B), or is rather a smooth crossover.

2) I would like to see more information about how parameter values were chosen in the model. To what extent are the parameter values trying to capture the specifics of the experiments, as opposed to being representative values which qualitatively reproduce the behavior observed in experiments? Specifically, the mechanical parameters are taken from Camley et al. (2014) in subsection “Full model”, but that reference appears to be a computational study rather than an experimental one – were the model parameters used in the earlier study connected to measurements of cell mechanics? If not, what is their source, and are they reasonable? The biochemical parameters are stated as being informed by experiments, but if so, how do the authors explain the large discrepancy between the oscillation period observed in stationary cells (~few minutes, Figure 3—figure supplement 2) and the oscillation period in the simulations (~20 s, Figure 2—figure supplement 5C)?

[Editors' note: further revisions were requested prior to acceptance, as described below.]

Thank you for resubmitting your work entitled "Plasticity of cell migration resulting from mechanochemical coupling" for further consideration at *eLife*. Your revised article has been favorably evaluated by Detlef Weigel as the Senior Editor, a Reviewing Editor, and three reviewers.

The manuscript has been improved but there are some remaining issues that need to be addressed before acceptance, as outlined below. I am copying the remaining comments of the reviewers below; these should not be too hard to address.

Reviewer #1:

In the revision, the authors have greatly addressed most of the concerns I raised in the previous review. In my opinion, the manuscript has really improved and it is now much clearer. Yet, I still have two major concerns about the comparison between the experimental and the theoretical migration-mode diagrams and about a check on the force-balance equation. To be clear, I do really appreciate the effort the authors made in building the experimental migration-mode but I think it would be best if they improved it as far as possible to show a straightforward comparison between theory and experiments.

– I understand that inferring the relationship between the "molecular effect" of latrunculin and the protrusive force by fluorescence analysis is currently difficult or not possible. Yet, I am missing why more quantitative comparisons cannot be tried whatsoever. For example, for a given value of the cell area (averaged over a bin for the experimental data), one could analyse the average velocity of the centre of mass varying latrunculin in the experiments and varying the protrusive force in the theory. I think that this would give a first hint on the relationship between the latrunculin concentration and the protrusive force and how robust this relationship is across different values of the cell area. The same could be carried out for the average curvature. Of course, this is only a suggestion: it could be highly possible that the outcome of this analysis is quite noisy and no relationship can be inferred. In this case, then, I would only focus on merging the three experimental migration-mode diagrams into one and using a thresholding/averaging procedure to make it more uniform and directly comparable to the theoretical one.

– As check, I went through the table of parameters and got a bit confused by the units of measurement of the parameters in the force balance equation.

For example, if I use the units of the friction coefficient reported in the table and plug them into the force balance equation (Equation 8 – Materials and methods), I obtain pN/μm^3^ on the LHS, which does not match the protrusive force term and the area conservation term on the RHS (note I assumed time in s, forces in pN, velocity in μm/s and dimensionless phase field). If I am not missing anything, I think that some of the forces should be rescaled by the disk surface to obtain the "point-by-point force balance" and Bs, the parameter constraining the cell area, should have dimensions pN/μm^2^. Could the authors clarify this point or tell me what I am missing? For the concentration module, should not the noise intensity have units μM^2^/s^2^ (instead of μM^2^/s)? The other units should be fine instead (but please have a final check).

---

## [Author Response]

The manuscript addresses the question of how cells switch between different migration modes by coupling mechanics and signalling. All three reviewers are generally positive about the paper. Yet they have certain concerns that would improve both the readability and strengthen the conclusions. We think it is instructive to present the reviewers comments to you, since they are for the most part convergent and they highlight what the reviewers appreciated about this work. Please note that we are aware that the answers to some of the comments are in the supplementary information, yet since the reviewers had a hard time finding them, it is likely that general readers would find this difficult as well.During the post-review discussion among the reviewers, additional concrete suggestions emerged:1) It would be helpful if the authors provided a precise definition of the amoeboid -> keratocyte transition in the model, and of the amoeboid phase in the experiments, and to move the relevant information (including Figure 2—figure supplement 2A and a similar panel for the amoeboid -> keratocyte transition in the model) to the main text and figures.

We have now provided a precise definition of the transitions between the different migration modes. This is discussed in the main text and in more detail in the new subsection of Materials and methods, “Identification of migration modes”. Also, we have added panels to Figure 2, which now show the transition from oscillatory to amoeboid-like cells (panel C), and from amoeboid-like to keratocyte-like cells (panel E).

2) Producing a "migration-mode diagram" from the existing experimental data to provide a more concrete comparison of model prediction to data would be really helpful. The reviewers are not asking for time consuming new experiments.

We thank the editors and reviewers for this suggestion. We have now added such a migration-mode diagram (new Figure 3F) which shows the percentage of cells using the different migration modes in the latrunculin concentration vs. cell area space. Importantly, the qualitative features of these phase diagrams agree very well with the computational phase diagram: large protrusive strength (corresponding to experiments carried out in the absence of latrunculin) results in large numbers of cells migrating in the keratocyte-like mode. Introducing latrunculin leads to a decrease in protrusive strength and the destabilization of some of the keratocyte-like cells into amoeboid-like cells. For even larger values of latrunculin, we find a larger percentage of cells that show oscillatory behavior, fully consistent with our numerical predictions.

Reviewer #1:[…] The manuscript is in general clear and well written, the literature well represented and the authors already addressed in the paper most of the major concerns I would have in principle. Yet, I list below the points I think would definitely improve the clarity and quality of the manuscript.1) The authors define a reaction-diffusion system made of an activator and inhibitor, respectively called A and R, which works in a relaxation oscillation regime. I do understand that this choice is made in order to observe alternated phases for the activator and the authors discuss their model reduction in the Discussion. Yet, I wonder whether the inhibitor presence is really necessary and a more coarse-grained model could actually reproduce the same results.

The coupling between traveling waves and cell mechanics is a crucial element in our model: waves propagate towards the membrane and continue to move as long as the membrane can “keep up” with the wave. Therefore, the ability of the biochemical model to generate propagating waves is necessary. To the best of our knowledge, a minimal model (either excitable or oscillatory) for robust traveling waves with a finite wavelength contains at least two components and these components can be thought of as the activator and inhibitor.

2) I have a general feeling that the model results could be better discussed in the main text. Most of the points about the variation of parameters are described in the supplementary materials and the reasoning behind the main text is hard to follow. To make the manuscript clearer, I would also move some of the supplementary figures in the (already existing) main figures by condensing/rearranging the panels and giving them proper titles (e.g., "Protrusive force effect", etc.), once the theory discussion in the main text has been expanded;

We thank the reviewer for this suggestion. We have substantially rearranged the text and have modified the figures. Specifically, we have added text to the Results section explaining some of the theoretical reasoning and have added 5 new panels to Figure 2. These panels display typical trajectories of our computational cells (panel B) and show the transitions from oscillatory to amoeboid-like cells and from amoeboid-like to keratocyte-like cells (panels C and E). Also, we have moved the supplementary figure which shows the effect of protrusion force on keratocyte-like cell shape to Figure 2D.

Furthermore, we have moved the supplementary panel showing the effects of parameter variations on the boundaries in the phase diagram to Figure 2G. With these modifications, we hope that the model results are easier to follow.

3) The authors model cell mechanics with a phase field method. The cell shape is described by an auxiliary field that defines the boundary of the cell changing as a function of mechano-chemical forces. I have two main questions about this point. First, I wonder why the authors did not use a more realistic continuous active-viscous shell description. In the context of the phase field model, any viscous dissipation of the membrane is completely neglected and I wonder whether this would only shift the migration-mode switch appearance or would strongly modify the results. Also, in general, the parameters in the phase-field model are quite coarse grained compared to those actually measurable experimentally (e.g., surface tensions). Can the authors clarify this point? Second, the coupling between the mechanical and signalling modules is taken to be strongly non-linear via a sigmoidal function of the activator. The authors analyse in depth the dependency of the cell migration modes on the intensity of this coupling. I do think it would be worth to investigate also whether and how changing the "Hill coefficient" could lead to motility switches in the model, as this would give more information about the mathematical description of the properties of the mechano-chemical coupling in this context and possible cooperative effects at the biochemical level.

We appreciate the comment of the reviewer and would like to stress that our goal was to create a model that was as simple as possible and able to capture the critical role of interplay between signaling waves, cell shape and mechanics. Adding fluid flow adds considerable complexity to our computational model and will be our next step. This addition can certainly affect the cell mechanics and might lead to cell moving speed different from Equation 4. However, we believe that it should not strongly modify the main results because migration mode transitions are mostly determined by matching waves to cell shape.

The biochemical parameters (including diffusion constants and reaction rates) are estimated to recapitulate the wave propagation measured in Gerhardt et al. (2014) and Arai et al. (2010). The membrane tension is the same as we have used in our earlier studies (Shao et al., 2010, Shao et al., 2012, Camley et al., 2014) and is estimated from an earlier experimental study (Simson et al., 1998). We now cite this study.

At the suggestion of the reviewer, we have investigated the effect of changing the Hill coefficient, n. These new simulation results are reported as a figure supplement (Figure 2—figure supplement 4C) where we plot the phase diagram of migration modes for different values of n. The simulations reveal that there is no qualitative difference for different values of n. In particular, for fixed cell size there is still a transition between oscillatory and amoeboid-like cells for a critical protrusion strength η_c,1_ and a transition from amoeboid-like to keratocyte-like cells for a larger protrusion strength (η_c,2_). Furthermore, the transition from oscillatory to amoeboid-like cells is independent of cell size for all values of n while η_c,2_ shows the same qualitative behavior as a function of cell size. The shifting of the η_c,1_ and η_c,2_ curves can be simply attributed to the fact that the integrated value of the sigmoidal function M(A) within the cells increases for increasing values of n.

4) To my knowledge, there are not universally accepted values for the bending modulus but just very wide ranges of parameters (if I am wrong, I would be very happy to have the reference precisely quantifying this). Could the authors describe whether they investigated the role of this parameter (I have looked into the supplementary materials but did not find any mention about it)? In principle, since bending prevents changes in the mean curvature, I would expect a smoothening or even a disappearance of the switch between the different migration modes.

We agree with the reviewer that there is experimental uncertainty about the bending modulus. We have followed up on the suggestion of the reviewer and have tested the effects of bending by increasing the modulus 2.5 fold (to 5 pN µm) and by setting it equal to zero. We found that the migration mode transitions are virtually unchanged. This is in line with our previous results (e.g., Camley et al., 2017) and, for simplicity, we have now removed the bending energy from the model.

5) The authors perform experiments on cells where their protrusive force is modulated by interfering with actin polymerisation. Their results all agree qualitatively with their model, i.e., decreasing the protrusive force destabilises keratocyte-like migration and allows the switch to the amoeboid-like migration. This has also effects on other quantities as front-back distance, speed and size as qualitatively found by the model. Yet, I do not see anywhere any quantitative comparison between model and experiments. Can the authors clarify this and show an experimental "migration-mode diagram"? A quantitative comparison would be very interesting as it would require in principle a more general definition of protrusive force in the model, as interfering with actin polymerisation alters all mechanical parameters such as tension and bending.

As we already mentioned above, we now include such a migration-mode diagram (Figure 3F) and discuss its results in the main text. This diagram agrees well with the computational predictions.

Reviewer #2:[…] The author's group has previously constructed similar models for cell boundary and motility. The biochemical model is based on a relaxation oscillator by Arai et al. (2010). Thus, this study likely combined the two existing models. Also, experimental evaluation is not conclusive. My major concerns are described below.

We thank the reviewer for their comments and suggestions. We would like to stress that our current model is more than a simple combination of two models. The reviewer correctly points out that we have presented studies of deforming and moving cells before (e.g., Camley et al., 2013, 2014, Shao et al., 2010 and 2012) and that the biochemical model is based on the important work by Arai et al. However, the novel and most important result of our current study is that coupling between a biochemical module and a mechanical module can produce different cell migration modes.

An important finding in this study is that the transition of migration modes depends on the protrusive strength (η) and cell size (r). But it is not clear what is the definition of each migration mode produced by the model. Previous work characterized the amoeboid- as well as keratocyte-like motility of Dictyostelium cells (Takagi et al., PLoS One, 2008: Asano et al., 2004). Does the model here recapitulate these features of real cell behaviors?

We thank the reviewer for pointing this out. We now explicitly include a definition of each migration mode in the main text as well as in a new subsection in the Materials and methods section. In simulations, we can define oscillatory cells as cells with a vanishing center of mass velocity while amoeboid-like and keratocyte-like cells can be distinguished using either the angles of the cell trajectory or the time-averaged curvature of this trajectory. Examples of these cell trajectories are now shown in Figure 2B. Since an amoeboid-like cell has an unstable polarity, its trajectory is more curved, resulting in a high average curvature. The trajectory of a keratocyte-like cell, on the other hand, is characterized by long and straight segments and thus much lower average curvature. As we now show in new panels Figure 2C and E, these definitions clearly capture the transitions between the different migration modes.

Also, I am not sure how the activator is maintained at the front of a keratocyte-like cell even after the inhibitor is produced.

The activator can be maintained because our activator-inhibitor system produces traveling wave. Viewed in a frame moving with the wave speed the front is stable and the inhibitor is always distributed at the back of the activator wave front.

The model was evaluated by adding latrunculin to reduce the protrusive strength in Figure 3 but there are some issues to be addressed. First, the criteria for each migration mode is missing in the manuscript.

We agree with the reviewer and have now added explicit definitions for the criteria for each migration mode.

Second, the keratocyte-like cells may differ between the model and the experiment. The model predicted that F-actin asymmetrically localize at the cellular front but the keratocyte-like cell shown in Figure 3A has F-actin at the entire cortex. Also, the model should be evaluated by drawing the phase-diagram in the (η,r) space of Dictyostelium cells shown in Figure 3A. The protrusive strength could be quantified by the fluorescence intensity of limE-GFP.

Please keep in mind that the variable A in our model does not represent F-actin. Rather, it can be thought of as an upstream component such as a PtdIns phosphate. Therefore, the distribution of A is not supposed to exactly correspond to the experimental distribution of F-actin. In addition, as can be seen by overlaying the fluorescence image of Figure 3A(v) with the corresponding DIC image, limE-GFP is not located at the back membrane of the keratocyte-like cell. This is further clarified in Author response image 1 in which we plot the fluorescence image of Figure 3A(v) and the cell outline in red (determined form the DIC image). This image shows that the maximum fluorescence intensity occurs away from the membrane. As shown in experiments (see, e.g., Asano et al., 2008) actin leads PIP3 at the front of the cell while it is trailing PIP3 at the back of the cell. This would imply that the distribution of PtdIns phosphates, including PIP3, will not extend to the membrane but is bounded by the actin distribution. We are currently pursuing further investigations to quantify the distributions of PIP3 and actin in migrating cells.

Reviewer 3, along with the other reviewers and the editors, suggests to construct an experimental phase diagram similar to Figure 2F. Our new panel (Figure 3F) provides such a phase diagram.

We thank the reviewer for the suggestion to quantify the protrusive strength using the fluorescence intensity. Unfortunately, our cells display a relatively large heterogeneity in limE-GFP expression levels which makes it not feasible to quantify the protrusive strength.

Third, the authors claim that the model predicts decreases in cell area size and front-back-distance of keratocyte-like cells after latrunculin treatment in the third paragraph of the subsection “Experimental Results” but I do not understand why. I thought that a decrease in protrusion strength increases front-back-distance shown in Figure 2—figure supplement 3A.

We thank the reviewer for pointing this out and we apologize for the confusion. Our simulations are carried out for constant cell size S_0_. This assumption, necessary to prevent cells from shrinking or expanding indefinitely and an immediate consequence of the fact that we are modeling a 2D cell, prevents us from making a direct comparison of the front-back distance in simulations and experiments. We have now removed the statement the reviewer was referring to. However, the fact that the cell size in the experiments decreases upon the introduction of latrunculin is consistent with our model. Larger cells have more “room” to initiate a new traveling wave and should therefore be less stable than smaller cells. Thus we can expect that smaller keratocyte-like cells will remain fan shaped after a latrunculin treatment while larger keratocyte-like cells will become amoeboid-like cells. In the main text, we now only have statement that the cell size will decrease after latrunculin treatment.

Fourth, in addition to the study by Miao et al. which is deeply discussed in this manuscript, Nishimura et al. previously published mathematical analysis for the transition of different migration modes (Nishimura, Ueda and Sasai, 2009; 2012). The authors should discuss these works as compared with the current model.

We thank the reviewer for pointing out these interesting references. We now compare these studies to our model in the Discussion section.

Fifth, the current model still can operate in the excitable regime of the PtdIns phosphate module in Figure 2—figure supplement 6. The recent study showed that multiple pathways including PtdIns phosphate and sGC regulate random cell motility through their distinct excitable properties (Tanabe et al., 2018). Moreover, the model presented by Arai et al. was revised by Fukushima et al. and a new feature of PtdIns phosphate dynamics via the mutual inhibition between PIP3 and PTEN was reported to ensure polarity formation. These points should be discussed so that the presented model could be more realistically considered.

We again thank the reviewer for pointing out these references. They are now included in the revised manuscript, where they are also discussed.

Reviewer #3:[…] I have two major concerns about the methodology which I would like to see addressed before publication.1) The nature of the transition from amoeboid-like to keratocyte-like behavior in the model is not adequately quantified. The transition is described as being from "unstable to stable polarity" and the procedure for classifying experimental trajectories is provided (subsection “Experiments”), but I did not see a rigorous definition of the phase in terms of quantities measured in simulations. How is η_c,2_ determined, and is there a sharp transition in some quantity akin to that seen in the cell speed for the stationary -> amoeboid transition (Figure 2—figure supplement 2A)? This information is needed to judge whether the amoeboid -> keratocyte transition is indeed a sharp one in the r-η plane (as suggested in the Abstract and in Figure 2B), or is rather a smooth crossover.

We thank Dr. Paulose for this comment, which was also brought up by the other reviewers. As we mentioned in our response to the editors and reviewer 1 and 2, we now include a description of our definitions of the three different migration modes. These definitions are based on the cell trajectories: in simulations, oscillatory cells are defined as cells with a vanishing center of mass velocity while amoeboid-like and keratocyte-like cells are defined based on either the average local curvature of the trajectory or based on the standard deviation of the angles between points along this trajectory. In experiments, oscillatory cells are determined by evaluating three criteria, as detailed in Materials and methods. These definitions are also used in the new panels Figure 2C and E which show the transition between oscillatory amoeboid-like cells and between amoeboid-like and keratocyte-like cells in simulations, respectively.

2) I would like to see more information about how parameter values were chosen in the model. To what extent are the parameter values trying to capture the specifics of the experiments, as opposed to being representative values which qualitatively reproduce the behavior observed in experiments? Specifically, the mechanical parameters are taken from Camley et al. (2014) in subsection “Full model”, but that reference appears to be a computational study rather than an experimental one – were the model parameters used in the earlier study connected to measurements of cell mechanics? If not, what is their source, and are they reasonable? The biochemical parameters are stated as being informed by experiments, but if so, how do the authors explain the large discrepancy between the oscillation period observed in stationary cells (~few minutes, Figure 3—figure supplement 2) and the oscillation period in the simulations (~20 s, Figure 2—figure supplement 5C)?

Dr. Paulose correctly points out that Camley et al. (2014) is a theoretical study. This study cites other studies which are used to estimate the tension. We apologize for this incomplete referencing and now include the original experimental reference (Simson et al., 1998).

Dr. Paulose is completely correct in noting that the oscillation period in the simulations (~20s) is much shorter (approximately by a factor of 10) than the one observed in experiments (~200s). The oscillation period is determined by the timescale of the inhibitor (τ) as well as the overall time scale of the model. To increase computational efficiency we have chosen τ to be small (10 s). However, rescaling this value by a factor of 2 while rescaling all time-dependent equations be a factor of 5 will result in an oscillation period that is 20x10=200s, consistent with the experimental value, and migration speeds and wavelengths that are comparable to experimental values.

[Editors' note: further revisions were requested prior to acceptance, as described below.]

The manuscript has been improved but there are some remaining issues that need to be addressed before acceptance, as outlined below. I am copying the remaining comments of the reviewers below; these should not be too hard to address.Reviewer #1:[…]– I understand that inferring the relationship between the "molecular effect" of latrunculin and the protrusive force by fluorescence analysis is currently difficult or not possible. Yet, I am missing why more quantitative comparisons cannot be tried whatsoever. For example, for a given value of the cell area (averaged over a bin for the experimental data), one could analyse the average velocity of the centre of mass varying latrunculin in the experiments and varying the protrusive force in the theory. I think that this would give a first hint on the relationship between the latrunculin concentration and the protrusive force and how robust this relationship is across different values of the cell area. The same could be carried out for the average curvature. Of course, this is only a suggestion: it could be highly possible that the outcome of this analysis is quite noisy and no relationship can be inferred. In this case, then, I would only focus on merging the three experimental migration-mode diagrams into one and using a thresholding/averaging procedure to make it more uniform and directly comparable to the theoretical one.

We thank the reviewer for this suggestion. We have quantified the average velocity of cells as a function of the latrunculin concentration for various cell areas (new Figure 3—figure supplement 1). This analysis shows that the average speed clearly decreases as the latrunculin concentration increases, indicating that adding latrunculin decreases the protrusive force.

Concerning the suggestion of the reviewer to merge the three experimental migration-mode diagrams into one we respectfully disagree. In our simulations, the migration mode is uniquely determined by initial conditions, resulting in a single phase diagram. In the experiments, on the other hand, we encounter a distribution of different migration modes for a given area and given latrunculin concentration. This makes it difficult to merge the diagrams and therefore we prefer to plot the percentage of cells for the different modes in three separate diagrams.

– As check, I went through the table of parameters and got a bit confused by the units of measurement of the parameters in the force balance equation.For example, if I use the units of the friction coefficient reported in the table and plug them into the force balance equation (Equation 8 – Materials and methods), I obtain pN/μm^3^ on the LHS, which does not match the protrusive force term and the area conservation term on the RHS (note I assumed time in s, forces in pN, velocity in μm/s and dimensionless phase field). If I am not missing anything, I think that some of the forces should be rescaled by the disk surface to obtain the "point-by-point force balance" and Bs, the parameter constraining the cell area, should have dimensions pN/μm^2^. Could the authors clarify this point or tell me what I am missing? For the concentration module, should not the noise intensity have units μM^2^/s^2^ (instead of μM^2^/s)? The other units should be fine instead (but please have a final check).

We thank the reviewer for pointing out the unit problem. We have corrected the units in the table as follows: tension γ, pNµm; ξ, pNs/µm; B_S_: pN/µm^2^. The noise intensity sigma has units of µM^2^µm^2^/s, because the delta function δ(t-t’) has units of 1/s while δ(r-r’) has units of 1/m^2^.